# Mesoporous Silica Nanoparticles as a Gene Delivery Platform for Cancer Therapy

**DOI:** 10.3390/pharmaceutics15051432

**Published:** 2023-05-08

**Authors:** Nisar Ul Khaliq, Juyeon Lee, Joohyeon Kim, Yejin Kim, Sohyeon Yu, Jisu Kim, Sangwoo Kim, Daekyung Sung, Hyungjun Kim

**Affiliations:** 1Department of Chemistry and Bioscience, Kumoh National Institute of Technology, 61 Daehak-ro, Gumi 39177, Republic of Korea; nisul151@kumoh.ac.kr (N.U.K.); juyeon831@kumoh.ac.kr (J.L.); luna3304@kumoh.ac.kr (J.K.); yejin1837@kumoh.ac.kr (Y.K.); 2Center for Bio-Healthcare Materials, Bio-Convergence Materials R&D Division, Korea Institute of Ceramic Engineering and Technology, 202 Osongsaengmyeong 1-ro, Osong-eup, Heungdeok-gu, Cheongju 28160, Republic of Korea; aelnlouis@yonsei.ac.kr (S.Y.); jisukim6650@yonsei.ac.kr (J.K.); tkddn365@yonsei.ac.kr (S.K.); 3Department of Chemical and Biomolecular Engineering, Yonsei University, 50 Yonsei-ro, Seodaemun-gu, Seoul 03722, Republic of Korea

**Keywords:** mesoporous silica nanoparticles, biocompatibility, surface modification, gene delivery, co-delivery, cancer therapy

## Abstract

Cancer remains a major global health challenge. Traditional chemotherapy often results in side effects and drug resistance, necessitating the development of alternative treatment strategies such as gene therapy. Mesoporous silica nanoparticles (MSNs) offer many advantages as a gene delivery carrier, including high loading capacity, controlled drug release, and easy surface functionalization. MSNs are biodegradable and biocompatible, making them promising candidates for drug delivery applications. Recent studies demonstrating the use of MSNs for the delivery of therapeutic nucleic acids to cancer cells have been reviewed, along with their potential as a tool for cancer therapy. The major challenges and future interventions of MSNs as gene delivery carriers for cancer therapy are discussed.

## 1. Introduction

Cancer remains a major health problem worldwide and its treatment remains challenging owing to the limitations of chemotherapy, which often result in side effects and drug resistance [1,2,3,4]. There is a need for alternative treatment strategies, such as gene therapy. Gene delivery is a promising approach for treating cancer by targeting specific genes and regulating their expression [5]. However, gene delivery requires an effective carrier that can protect the therapeutic genes from degradation, facilitate cellular uptake, and promote the release of the therapeutic genes within the cells [6,7].

Several gene delivery systems have been developed, including organic and inorganic carriers [8,9,10]. Organic carriers such as liposomes, polymeric nanoparticles, and dendrimers have been widely studied for gene delivery because of their biocompatibility and ability to encapsulate a wide range of therapeutic molecules [11,12]. However, their low stability and limited ability to control drug release have led to the development of inorganic carriers, such as mesoporous silica nanoparticles (MSNs) [13].

MSNs are a class of inorganic materials with unique properties, including a high surface area, tunable pore size, and biocompatibility, which make them attractive candidates for drug delivery applications [14,15]. MSNs have been extensively examined as drug delivery platforms for cancer therapy, and their use as gene delivery carriers has gained increasing attention in recent years [16,17]. MSNs offer several advantages as gene-delivery carriers, including high loading capacity, controlled drug release, and easy surface functionalization [18,19]. Furthermore, MSNs are biodegradable, and their degradation products, such as silicic acid, are naturally occurring substances that are safe for human use [20].

In this review, we focus on the use of MSNs as a gene delivery carrier for cancer therapy. We will discuss the required properties of MSNs that make them suitable for gene delivery, biocompatibility, biodegradability, and safe. We will also review recent studies that have demonstrated the use of MSNs for the delivery of therapeutic nucleic acids to cancer cells and their potential as a tool for cancer therapy. Finally, we highlighted the challenges and future perspectives of MSNs as a gene delivery carrier for cancer therapy.

## 2. Mesoporous Silica Nanoparticles

### 2.1. Synthesis of Mesoporous Silica Nanoparticles

MSNs are typically synthesized via the hydrolysis and condensation of silica precursors in the presence of surfactants or other templating agents that can stabilize and direct the growth of the nanoparticle structure. The choice of the synthesis method and conditions can affect the size, shape, and pore structure of the resulting MSNs, which in turn can influence their properties and applications [14]. The Stöber method is one of the most widely used methods for MSNs synthesis, which involves the hydrolysis and condensation of tetraethylorthosilicate (TEOS) in the presence of ammonia and ethanol as a catalyst, and water as a hydrolyzing agent (Figure 1a) [21]. The surfactant-templating approach, on the other hand, involves the use of surfactants as structure-directing agents to control the size and shape of the MSNs. This approach can be further divided into various subcategories, such as cationic, anionic, nonionic, and neutral templating methods, depending on the nature of the surfactant used [22]. Another commonly used method is the sol–gel process, which involves the hydrolysis and condensation of metal alkoxides or other precursors in solution or in a gel matrix (Figure 1b) [23]. Despite advances in MSNs synthesis, achieving high uniformity and reproducibility remains a challenge, and further optimization and characterization of the synthesis methods are needed to fully realize the potential of MSNs for various applications.

### 2.2. Stability of Mesoporous Silica Nanoparticles

Silica is a highly stable material, but its stability can be affected by various factors, including the presence of amines in aqueous media. Amines can act as Lewis bases, which means they have lone pairs of electrons that can coordinate with Lewis acids, such as the silicon atoms in silica [24]. Silica surfaces are naturally prone to nucleophilic attack in aqueous solutions, which can cause erosion of their external structure and induce premature damage. The effect of amine-promoted dissolution on MSNs can be either favorable or detrimental depending on the surface groups disposed of onto their surface [25]. To preserve the integrity of MSNs against amine-promoted dissolution in aqueous media, several methods have been developed, including surface modification and functionalization. For instance, one approach to preserving the integrity of MSNs against amine-promoted dissolution in aqueous media is to functionalize the nanoparticles with amine groups. This can be achieved by using 3-aminopropyl (triethoxy) silane (APTES), which is commonly used as a precursor for conjugating biomolecules to the surface of MSNs [26]. Another approach is surface functionalization based on zwitterions, which are neutral compounds with formal electrical charges of opposite signs. This approach has been demonstrated to prevent fouling and aggregation of nanoparticles in biological media [27].

### 2.3. Biocompatibility of Mesoporous Silica Nanoparticles

The biocompatibility of MSNs is a crucial consideration for their use in various biomedical applications, such as drug delivery, imaging, and sensing. MSNs are promising because of their unique physical and chemical properties, including their large surface area, tunable pore size, and surface chemistry [28]. However, there is a need to thoroughly examine the biocompatibility, in terms of safety, in vivo. In general, studies have reported that MSNs exhibit low toxicity and minimal adverse effects in vitro and in vivo, because of the silica composition, which is a naturally occurring and biocompatible material [20,29]. MSNs have a small size, which enables them to escape rapid clearance by the reticuloendothelial system (RES), and tunable surface chemistry, which can promote their uptake by target cells and tissues [30]. MSNs can be functionalized with a wide variety of biomolecules, such as proteins, peptides, and nucleic acids, to enhance their biocompatibility and target specificity [31,32]. However, the biocompatibility of MSNs can be influenced by various factors, such as their size, shape, surface chemistry, and surface charge [33]. MSNs with large sizes or highly charged surfaces may cause toxicity or immune responses, and their long-term biodegradation and clearance from the body are still under investigation. In vivo biocompatibility of MSNs may also be affected by their route of administration, such as intravenous, intraperitoneal, or intratumoral, and their interaction with different physiological environments, such as blood, lymph, or tumor microenvironment [34,35]. Therefore, careful evaluation of the biocompatibility of MSNs is critical for their safe and effective use in biomedical applications. Pre-clinical studies using appropriate animal models are necessary to investigate drug safety, efficacy, pharmacokinetics, and bio-distribution. In addition, suitable methods for the characterization and quality control of MSNs need to be established to ensure their reproducibility and consistency.

### 2.4. Size and Morphology of Mesoporous Silica Nanoparticles

MSNs have gained significant attention due to the ease of tailoring their properties, particularly in morphology, particle size, and pore size. For drug delivery systems, nanoparticle size is a critical factor and must fall within the range of 10 nm to 400 nm [36]. In nanomedicine, the issue of size effect on cell uptake of nanoparticles has become a significant concern. Particle size plays an essential role in designing appropriate cell-tracking and drug carrier nanoparticle systems, as it affects the mechanism and rate of cellular uptake and the ability to permeate through tissue. Understanding the effect of particle size on cellular uptake is crucial for all applications of nanoparticles in nanomedicine. Lu et al. have demonstrated a simple method for controlling the size of well-ordered and dispersed MSNs ranging from 30 to 280 nm by adjusting the pH of the reaction solution [37]. Furthermore, we have investigated the impact of particle size on the uptake of fluorescein isothiocyanate (FITC)-MSNs by HeLa cells and found that uptake is particle size dependent, with the maximum uptake occurring at a nanoparticle size of 50 nm. These findings suggest that MSNs with a diameter of 50 nm may be the most suitable candidate for carrying out further biological applications.

Additionally, particle shape is a crucial factor in determining the interaction of nanoparticles with cells and their systemic distribution. Investigating the effect of different-shaped particles on cellular behavior can assist in the design of nanoscale delivery systems and open up new applications for nanoparticles. For instance, Huang et al. investigated the effect of particle shape on the cellular uptake of MSNs functionalized with FITC and rhodamine B isothiocyanate (RITC) for imaging and quantification of MSN uptake [38]. The results showed that the number of particles internalized by cells strongly depended on particle shape. Long rod-shaped MSNs were more easily internalized by cells than short rod-shaped and spherical MSNs. Moreover, the uptake rate of long rod-shaped MSNs was more rapid than short rod-shaped MSNs. In contrast, fewer spherical MSNs penetrated the cells than short rod-shaped and long rod-shaped MSNs, indicating that the cellular uptake of short rod-shaped and long rod-shaped MSNs was faster than spherical MSNs. This study highlights the importance of particle shape in cellular uptake and provides insights for the design of more efficient nanoscale delivery systems.

### 2.5. Mesoporous Silica Nanoparticles for Gene Delivery

Gene therapy has emerged as a promising approach for treating various genetic and acquired diseases, by introducing exogenous genes or modulating the expression of endogenous genes [39,40]. MSNs have shown great potential as gene delivery vehicles, because of their large surface area, tunable pore size, and ease of functionalization. The porous structure of MSNs allows for the encapsulation or adsorption of nucleic acids, such as plasmid DNA or siRNA, within the pores or on the surface of the particles, while protecting nucleic acids from degradation by nucleases or other enzymes [41]. The uniform pore size and distribution of MSNs can also provide a high degree of control over the release of nucleic acids, either through passive diffusion or by applying external stimuli, such as pH or temperature [42,43]. The surface of MSNs can be functionalized with various ligands or targeting moieties, such as cationic polymers or peptides, to enhance their cellular uptake and transfection efficiency [44,45]. MSNs have shown high transfection efficiency in various cell types, including cancer cells, stem cells, and immune cells, with low cytotoxicity and minimal immune response. MSNs have also been used for in vivo gene delivery, with promising results in animal models of cancer, inflammation, and genetic disorders [46,47,48,49]. In conclusion, the synthesis and biocompatibility of MSNs as well as their potential as gene delivery vehicles. MSNs can be synthesized using various methods, and the choice of method affects their properties and applications. Although MSNs have shown low toxicity and minimal adverse effects in vitro and in vivo, their biocompatibility needs thorough evaluation, particularly in vivo, to ensure their safe and effective use in nanomedicine applications. MSNs have great potential as gene delivery vehicles due to their large surface area, tunable pore size, and ease of functionalization, resulting in high transfection efficiency with low cytotoxicity and minimal immune response in various cell types. In vivo studies have also shown promising results in various diseases.

## 3. Mesoporous Silica Nanoparticles-Based Gene Delivery for Cancer Therapy

Surface modification of MSNs can greatly affect their required physicochemical properties, which can impact their efficacy as gene delivery carriers. One of the most commonly used approaches to modify the surface of MSNs for gene delivery is the functionalization of the silica surface with positively charged groups, such as amines, quaternary ammonium, or guanidine groups, which can facilitate the binding of negatively charged DNA or siRNA molecules through electrostatic interactions. Another approach involves coating MSNs with cationic polymers which can provide additional positive charge for nucleic acid binding, as well as enhanced endosomal escape through the proton sponge effect. The following sections have been selected to highlight examples of functionalization strategies for MSNs used for gene delivery in cancer therapy.

### 3.1. Surface-Functionalized MSNs for Gene Delivery

MSNs possess surface silanol groups that can be ionized, leading to a negative charge on their surface [50,51]. This negative charge can reduce the ability of MSNs to form a complex with negatively charged nucleic acids. Accordingly, surface functionalization has become a widely used approach to introduce a net positive charge onto the surface of MSNs, enabling them to effectively interact with negatively charged nucleic acids. Positively charged functionalized MSNs have been extensively studied as promising gene delivery carriers. Among the surface functionalization methods, the use of 3-aminopropyltriethoxysilane (APTES) has gained considerable attention owing to its excellent biocompatibility, high stability, and the ability to introduce primary amine groups on the surface of MSNs. The primary amine groups of the APTES-functionalized MSNs interact with negatively charged nucleic acids, forming stable electrostatic complexes, which can effectively protect and deliver the nucleic acids to the target site. For example, Na et al. reported a highly efficient siRNA delivery system based on MSNs with larger pores, specifically with a diameter of 23 nm (MSN23) [52]. In contrast to many other MSNs that have small pore sizes (<5 nm in diameter), the larger pore size of MSN23 allows for efficient loading of siRNA molecules, which are approximately 7.5 nm in length and 2 nm in diameter. MSN23 was prepared by a simple procedure that involved the expansion of small pores, approximately 2 nm in diameter, of conventional MSNs through treatment with trimethylbenzene [53]. The siRNAs were easily loaded into the pores of MSN23 by mixing them with the amine-functionalized MSN23, allowing for efficient cellular delivery to induce vascular endothelial growth factor (VEGF) gene knockdown in MDA-MB-231 xenografts in vivo while maintaining their chemical integrity in the presence of serum proteins.

The surface coating of MSNs is often required to enhance the delivery efficacy of nucleic acids. This is because nucleic acids have been found to be mostly immobilized or adsorbed on the outer surface of MSNs instead of being encapsulated inside MSN pores, mainly due to the relatively small pore size of MSNs, which limits their capacity to accommodate such large biomacromolecules. As a result, surface-bound nucleic acids are susceptible to enzyme-mediated degradation in vivo, which can reduce the efficacy of the gene delivery system [54,55]. Heidari et al. utilized chitosan as a surface coating material for the multiple drug-resistant protein-1 (MDR1) siRNA-loaded NH_2_-MSNs [56]. Chitosan was used to coat the MDR1 siRNA as a protective layer to address concerns about RNA degradation by nucleases in physiological fluids, which can occur with physical adsorption in gene delivery systems. The chitosan coating on the surface of the MDR1 siRNA-NH_2_-MSNs was found to significantly improve MDR1 siRNA protection against enzyme activity, compared to the naked siRNA-NH_2_-MSNs. These findings suggest that the incorporation of chitosan as a surface coating material can effectively enhance MDR1 siRNA stability and protection and decrease MDR1 transcription and translation in vitro in HeLa-RDB and EPG85.257-RDB cells. Xiong et al. developed a polo-like kinase 1 (PLK1) siRNA delivery system that utilizes amine-functionalized magnetic core–shell silica nanoparticles (MNC) with 12 nm radial large mesopores [57]. They applied an acid-labile coating to this carrier, which was formed through the pH-dependent complexation between tannic acid (TA), a polyphenolic ligand, and metal ions [58]. The surface coating was prepared in a simple and efficient manner by mixing solutions containing Al^3+^ ions and TA with the PLK1 siRNA-loaded MNC (MNC@siRNA) and adjusting the reaction pH. Compared to previous carrier coating strategies using lipid bilayer or cationic polymers [59], this TA coating exhibited high biocompatibility, availability to various substrates, and a facile one-step preparation process [60,61]. In addition, the coating was pH-dependent and could be easily decomposed under acidic conditions. The acid-labile surface coating MNC@siRNA, composed of tannic acid, serves a dual purpose as both a pore-capping agent to protect PLK1 siRNA and as a pH-responsive switch for on-demand intracellular release of the PLK1 siRNA in the human osteosarcoma KHOS cell line in vitro. Polydopamine, a polyphenol-based material similar to tannic acid, has also been used as a surface coating. Kim et al. utilized polydopamine to coat MSNs loaded with PDL1 siRNA, which recruited serum albumin, utilized caveolae-mediated endocytosis for tumor cell entry, and effectively silenced target genes (Figure 2) [62]. The MSNs core was removed to produce a hollow capsule, called Nanosac, to avoid issues related to cationic carriers and exploit the advantages of a flexible system. This system, serving as a carrier of siRNA targeting PD-L1, significantly reduced CT26 tumor growth through immune checkpoint blockade.

The CRISPR-associated protein 9 (Cas9) system is a recently developed technology capable of efficient and precise gene editing, which has the potential to permanently alter target genes and offer effective treatments for cancer [63]. Liu et al. have developed a virus-like nanoparticle (VLN) with a core–shell structure to co-deliver CRISPR/Cas9 system and small molecule drugs for effective cancer treatment [64]. The MSN-based core of VLN is loaded with small molecule drugs and CRISPR/Cas9 system and encapsulated with a lipid shell. By loading VLN with a single guide RNA (sgRNA) targeting programmed death-ligand 1 and axitinib, multiple immunosuppressive pathways were disrupted, and melanoma growth was suppressed in vivo. VLN can co-deliver various combinations of sgRNAs and small molecule drugs to tumors, indicating its potential as a platform for advanced combination therapies against malignant tumors.

### 3.2. Polycation-Coated MSNs for Gene Delivery

Polycation-coated MSNs have been explored as gene delivery carriers. One commonly used example of polycation is polyethyleneimine (PEI). PEI has been shown to effectively condense and protect DNA, facilitate endosomal escape, and promote gene expression. The positive charge of PEI allows it to interact with the negatively charged DNA, leading to the formation of compact complexes that can efficiently deliver genes to the cells.

A layer-by-layer approach has been employed to functionalize MSNs with polycations. This involves the first coating of the MSNs with a negatively charged layer, such as silica or poly (acrylic acid), followed by the deposition of the polycation layer. The resulting polycation-coated MSNs have been shown to be effective for gene delivery, with high transfection efficiency and low cytotoxicity [55,65,66,67]. However, despite their potential as gene delivery carriers, some challenges remain associated with polycation-coated MSNs. One major challenge is their tendency to aggregate, which can affect their stability and ultimately their gene delivery efficiency [68,69]. Additionally, the high positive charge of polycations such as PEI can lead to non-specific interactions with cell membranes, leading to toxicity issues [70].

To address these limitations, several research groups have reported surface modification of polycation-coated MSNs for cancer gene delivery. Shen et al. developed a siRNA carrier using cyclodextrin-grafted polyethylenimine (CP)-functionalized MSNs (CP-MSNs) [71]. CP-MSNs form an electrostatic bond with pyruvate kinase M2 isoform (PKM2) siRNA and eliminate the charge-induced toxicity of PEI through the use of cyclodextrin. CP-MSNs were assessed for their cellular internalization, gene silencing capability, and anticancer activity in MDA-MB-231 human breast cancer cells, as well as their tumor accumulation and in vivo gene silencing efficiency using an orthotropic mouse model of MDA-MB-231 breast cancer. As another example, Chen et al. presented a VEGF siRNA delivery based on magnetic MSNs (M-MSNs) functionalized with PEI and fusogenic peptide (KALA), which effectively silenced target genes both in vitro and in vivo [72]. This delivery system involved encapsulation of VEGF siRNA in M-MSNs mesopores, coating of PEI on VEGF siRNA-loaded M-MSNs’ surface, and conjugation of KALA peptides. The resulting vehicles, with high VEGF siRNA protection and low cytotoxicity, were readily internalized into cells, escaped from the endolysosomes, and released the VEGF siRNA into the cytoplasm, leading to excellent RNAi efficiency in tumor cells for enhanced VEGF knockdown in vitro and in vivo.

MSNs can also serve as gene carriers by loading DNA into their mesopores. However, the pore size of most MSNs used for drug delivery is too small to accommodate large DNA molecules, restricting nucleic acids to the nanoparticle surface. To overcome this challenge, Gao et al. developed MSNs with large pores (approximately 20 nm) that enable the loading and release of plasmids [73]. In a later study, Zhu et al. created organically functionalized and PEI-coated hollow MSNs with large pores, which were able to deliver a plasmid encoding hepatocyte growth factor (HGF) into bone marrow-derived mesenchymal stem cells, showing promise for regenerative medicine applications [74]. In summary, surface modification of MSNs is crucial for effective gene delivery as the presence of a negative charge on MSNs can hinder their interaction with negatively charged nucleic acids. Functionalization of MSNs with positively charged groups, such as amines or quaternary ammonium, or coating with cationic polymers such as PEI, can introduce a net positive charge and facilitate gene delivery through electrostatic interactions and endosomal escape. APTES is a commonly used surface functionalization method for MSNs, while PEI-coated MSNs have shown promising results in condensing and protecting DNA, promoting gene expression, and delivering genes efficiently to cells. These functionalization strategies hold the potential for gene delivery in cancer therapy.

## 4. Mesoporous Silica Nanoparticles-Based Co-Delivery System

Cancer therapy often involves multiple approaches to increase its efficacy. In addition to traditional chemotherapy, gene therapy has been recognized as a promising treatment for cancer because of its ability to target the underlying genetic causes of the disease. As a result, there has been a surge of interest in creating co-delivery systems for cancer therapy based on MSNs, which combine both chemotherapy and gene therapy. These MSN-based systems have the potential to enhance the therapeutic effects of both approaches, providing a more comprehensive and efficient cancer treatment. MSNs have enabled the delivery of nucleic acids along with drugs to improve drug effectiveness and maximize cellular uptake. Traditional methods of drug/gene delivery involve injecting a gene-carrying virus and administering a therapeutic drug directly into the tumor, which often results in low efficacy owing to pharmacokinetic differences between the small molecule drug and nucleic acid [75,76]. The large surface area, well-established surface chemistry, and porous structures of MSNs make them highly advantageous for co-delivery systems in cancer therapy. These features enable the encapsulation of both drugs and nucleic acids, thereby enhancing efficacy and improving drug delivery to cancer cells. The following examples demonstrate the diverse therapeutic strategies that can be achieved through the co-delivery of small molecule drugs and therapeutic nucleic acids using MSNs for oncology. (Table 1) The gene/drug combination investigation can be categorized based on the intended therapeutic effect of the nucleic acids in the combination. These include the prevention of resistance to chemotherapeutic drugs via multiple drug resistance mechanisms, the provision of anticancer therapeutic efficacy through angiogenesis inhibition, and the induction of cancer cell death via a distinct mechanism from that of the small molecule drug.

### 4.1. Prevention of Multiple Drug Resistance

Multiple drug resistance (MDR) remains a significant challenge in conventional anticancer therapy because of pump-mediated and non-pump forms of resistance [77]. The former involves the production of pumps by cancer cells that expel chemotherapeutic drugs, such as P-glycoprotein (Pgp), leading to higher drug concentrations required for treatment. On the other hand, non-pump resistance is driven by the overexpression of antiapoptotic proteins (e.g., B cell lymphoma 2 [Bcl-2]) that prevent apoptosis in cancer cells [78,79].

To inhibit Pgp, Sun et al. developed a unique core–shell hierarchical mesoporous silica/organosilica nanosystem with large and small mesopores in the shell and core, respectively, facilitating independent encapsulation of siRNA and doxorubicin (Figure 3) [80]. The organosilica shell with disulfide bonds can encapsulate large siRNA molecules, whereas the silica core provides reservoirs for small drug molecules. As the disulfide bonds within the shell break up in the tumor microenvironment, large siRNAs are initially released to suppress Pgp expression for pre-inhibition of MDR, followed by the release of the small drug molecules to exert an efficient therapeutic effect. This system offers significant advantages over traditional small or large mesoporous nanosystems for drug co-delivery. Meng et al. designed a dual delivery system using PEI-coated MSNs to simultaneously deliver doxorubicin and a Pgp-inhibiting siRNA [67]. This co-delivery approach resulted in increased cytotoxicity compared to particles delivering only the drug, as observed in a multidrug-resistant cell line. In a subsequent study, the author improved the system by optimizing the ratio of Pgp siRNA to doxorubicin and using PEGylated PEI-coated MSNs in an in vivo model of multidrug-resistant breast cancer [81]. The improved system showed synergistic inhibition of tumor growth and significant Pgp knockdown at heterogeneous tumor sites. Using a similar approach, Wang et al. designed a dual delivery system based on PEI-coated MSNs for treating oral squamous carcinoma [82]. The positively charged surface of the MSNs facilitated the co-delivery of MDR1 siRNA and doxorubicin. In vivo studies demonstrated that this system significantly reduced the tumor size (81.64% decrease after 28 days) and slowed tumor growth rate compared to the control group. Overall, the system showed enhanced efficacy in treating MDR cancer compared to doxorubicin monotherapy.

For the prevention of Bcl-2, Chen et al. were the first to utilize MSNs coated with G2 amine-terminated polyamidoamine dendrimers to simultaneously deliver doxorubicin and Bcl-2-targeted siRNA into MDR A2780/AD human ovarian cancer cells for enhanced chemotherapy efficacy [55]. Their study showed that the co-delivery of doxorubicin and Bcl-2 siRNA effectively silenced Bcl-2 mRNA, significantly suppressed non-pump resistance, and substantially enhanced the anticancer action of doxorubicin. Ma et al. developed a redox-responsive drug/siRNA co-delivery system using mesoporous silica nanoparticles for simultaneous in vitro and in vivo delivery [83]. To achieve controlled release, the author introduced ethylenediamine-modified β-cyclodextrin rings on the nanoparticle surface, which acted as smart nano-gates to block drug molecules within the mesopores and allow complexation with siRNA through electrostatic interactions. The system demonstrated redox-triggered disulfide bond cleavage for controlled drug/siRNA release in an intracellular environment.

The use of nanoparticles for targeted delivery of anti-tumor agents to tumor cells or tissues is desirable to reduce drug toxicity and improve therapeutic efficacy. A tumor-targeted MSN-based drug delivery system was developed for the inhalation treatment of lung cancer [84]. This system effectively delivered anticancer drugs (doxorubicin and cisplatin) and two types of siRNA targeted to multidrug resistance-associated protein 1 (MRP1) and Bcl-2 mRNA for suppression of the pump and non-pump cellular resistance in non-small cell lung carcinoma, respectively. The MSNs were targeted to cancer cells by conjugating LHRH peptide on the surface of MSNs via a poly (ethylene glycol) spacer. Folic acid has been widely used as an effective target ligand for targeted cancer therapy [85]. Ma et al. developed pH-responsive hollow mesoporous silica nanoparticles (HMSNPs)-based drug/siRNA co-delivery vehicles that can deliver doxorubicin and siRNA against Bcl-2 protein simultaneously to targeted cancer cells [86]. The hollow structure of HMSNPs improves the loading of anticancer drugs and enhances siRNA binding to certain cancer cells. The cancer-targeting effect of the nanoparticle carriers was demonstrated through folic acid receptor-mediated endocytosis in folic acid receptor-positive HeLa cells. Effective silencing of the anti-apoptotic protein Bcl-2 in HeLa cells resulted in enhanced therapeutic efficacy. Similarly, a co-delivery system based on MSNs was developed for the targeted and simultaneous delivery of an anticancer drug and siRNA into cancer cells [87]. The MSNs were modified with PEI-poly-l-lysine (PLL) via cleavable disulfide bonds and then conjugated with polyethylene glycol (PEG) linked to folate to target the folate receptor.

### 4.2. Inhibition of Angiogenesis

Cancer cells often produce various cytokines, including VEGF, to trigger angiogenesis and facilitate their rapid growth by increasing the nutrient and oxygen supply [88,89]. This simultaneous promotion of angiogenesis and proliferation is a hallmark of most cancers [90,91], making them difficult to treat with single-target therapies. To address this challenge, cocktail therapy has been developed, which combines cytotoxic chemotherapeutic agents with anti-angiogenic genes to effectively target cancer cells and prevent angiogenesis. Moreover, combination therapies are designed with patient heterogeneity in mind, rather than solely focusing on individual biological processes. By utilizing combination therapies, cancer treatment can be optimized for the specific needs of each patient. Han et al. developed multi-layered nanocomplexes (MLNs) that provide intelligent co-delivery of doxorubicin and VEGF siRNA (Figure 4) [92]. The MLNs were formed through electrostatic self-assembly of TAT peptide-modified mesoporous silica nanoparticles (TAT-MSNs) as a cationic core for doxorubicin loading, an anionic inner layer composed of poly (allylamine hydrochloride)-citraconic anhydride (PAH-Cit), and a cationic outer layer of galactose-modified trimethyl chitosan–cysteine (GTC) conjugate for siRNA encapsulation. The MLNs selectively and effectively delivered doxorubicin and siRNA to the nucleus and cytosol, respectively, by intelligently overcoming various physiological and biological barriers in the tumor zone in response to certain stimuli or phenotypes. Li et al. utilized folate receptor-targeted magnetic mesoporous silica nanoparticles to deliver a combination therapy of small hairpin RNA against VEGF (VEGF shRNA) and doxorubicin [93]. The nanocomplexes exhibited high doxorubicin loading capacity, efficient siRNA protection, and robust gene silencing activity of VEGF shRNA, resulting in effective inhibition of endothelial cell migration. Zheng et al. developed MSNs carriers conjugated with folate for the co-delivery of ursolic acid (UA) and VEGF siRNA [94]. UA has various pharmacological properties, including anti-tumor [95,96], anti-inflammatory [97], liver-protective [98], anti-atherosclerotic [99], anti-epileptic [100], and anti-diabetic [101]. The MSNs surface was modified with a covalently linked folic acid (FA) molecule via an acid-labile amide bond. UA was encapsulated within the MSNs-FA carrier through noncovalent interactions, whereas VEGF siRNA was loaded into the carrier via electrostatic interaction. The MSNs-FA nanocarrier co-loaded with UA and VEGF siRNA was expected to enhance the solubility and bioavailability of UA and improve the stability of VEGF siRNA. Zheng et al. developed an asialoglycoprotein receptor (ASGPR)-targeted nanodrug delivery system using MSNs nanocarrier for co-delivery of sorafenib (SO) and VEGF-targeted siRNA to hepatocellular carcinoma. [102] Sorafenib is a multi-kinase inhibitor that targets many growth factor receptors, inhibiting tumor growth and neoangiogenesis [103]. Lactobionic acid (LA), derived from lactose oxidation, was used as a targeting ligand because of its ability to specifically target human hepatocellular carcinoma [104,105]. The MSNs-LA delivery system induced S cell cycle arrest, enhanced cytotoxicity and tumor targeting of SO and VEGF siRNA, and improved VEGF siRNA transfection efficiency in ASGPR-overexpressing Huh7 cells, as demonstrated by in vitro testing.

### 4.3. Induction of Cancer Cell Death

A strategy for enhancing the efficacy of cancer therapy involves co-delivering nucleic acids and small molecule drugs to induce cell death through multiple cytotoxic mechanisms. One approach is to combine a chemotherapeutic drug with a nucleic acid that can independently induce cell death, leading to a synergistic effect. One potential target is the signal transducer and activator of transcription 3 (STAT3), a transcription factor that plays a crucial role in cell division, proliferation, and survival [106]. STAT3 activation is frequently observed in breast cancer cells, and inhibiting STAT3 using siRNA offers a promising approach for inhibiting human tumor cells [107]. Shakeran et al. developed and tested an in vitro nanoparticle system for the co-delivery of methotrexate (MTX) and STAT3 siRNA using MSNs functionalized with chitosan via covalent grafting mediated by APTES and glutaraldehyde as the linker [108]. The amine-rich surface coating provided by APTES and grafted chitosan enabled electrostatic loading of both MTX and siRNA. In vitro studies demonstrated effective co-delivery in MCF7 cells, inhibition of cellular division and proliferation, and decreased STAT3 expression at the mRNA and protein levels. In addition to it, the p53 tumor suppressor protein plays a vital role in promoting apoptosis in damaged cells or those with significantly compromised genetic material. However, the p53 gene is frequently mutated in various human tumors. As a result, delivering functional p53 to target cancer cells has been proposed as a promising therapeutic strategy with the potential for future translation [109]. Li et al. developed and tested a co-delivery system based on hollow mesoporous silica nanospheres for delivering a proteasome inhibitor, bortezomib (BTZ), and the tumor suppressor gene p53 to treat p53-mutant human non-small cell lung cancer (NSCLC) [110]. The system showed improved therapy efficacy for p53-mutant NSCLC in comparison to the treatment with BTZ alone. Rong et al. also used the same type of combination for the synthesis of the H2A-hybrid silica-coating upconversion nanoparticles and effectively utilized them for co-delivering the p53 gene and bortezomib [111]. This carrier exhibits low cytotoxicity and hemolysis activity, high payload capacity for genes and drugs, as well as high gene transfection efficiency. In conclusion, MSNs have shown potential as co-delivery systems for cancer therapy, enabling the encapsulation of both drugs and nucleic acids to enhance efficacy and improve drug delivery to cancer cells. MSNs have advantages over traditional methods of drug/gene delivery, including the ability to deliver nucleic acids along with drugs to improve drug effectiveness and maximize cellular uptake. The co-delivery of small molecule drugs and therapeutic nucleic acids using MSNs can achieve diverse therapeutic strategies in oncology, such as prevention of resistance to chemotherapeutic drugs via multiple drug resistance mechanisms, anticancer therapeutic efficacy through angiogenesis inhibition, and induction of cancer cell death via a distinct mechanism from that of the small molecule drug. Cocktail therapy, which combines cytotoxic chemotherapeutic agents with anti-angiogenic genes, has been developed to effectively target cancer cells and prevent angiogenesis, overcoming the challenge of simultaneous promotion of angiogenesis and proliferation in most cancers.

**Table 1 pharmaceutics-15-01432-t001:** Co-delivery of gene and small molecule based on MSN for cancer therapy.

Gene	Small Molecule	Target of Gene	Cancer Cell	Reference
Pgp siRNA	Doxorubicin	MDR prevention	MCF-7/ADR	[80]
Pgp siRNA	Doxorubicin	MDR prevention	MCF-7/MDR	[81]
MDR1 siRNA	Doxorubicin	MDR prevention	KBV	[82]
Bcl-2 siRNA	Doxorubicin	MDR prevention	A2780/AD	[55]
Bcl-2 siRNA	Doxorubicin	MDR prevention	HeLa	[83]
Bcl-2 siRNA, MRP1 siRNA	Doxorubicin, Cisplatin	MDR prevention	A549	[84]
Bcl-2 siRNA	Doxorubicin	MDR prevention	MCF-7, HeLa	[86]
Bcl-2 siRNA	Doxorubicin	MDR prevention	MDA-MB-231	[87]
VEGF siRNA	Doxorubicin	Anti-angiogenesis	QGY-7703	[92]
VEGF shRNA	Doxorubicin	Anti-angiogenesis	HeLa	[93]
VEGF siRNA	Ursolic acid	Anti-angiogenesis	HeLa	[94]
VEGF siRNA	Sorafenib	Anti-angiogenesis	Huh7	[102]
STAT3 siRNA	Methotrexate	Cancer cell death	MCF7	[108]
p53 pDNA	Bortezomib	Cancer cell death	NCI-H1299	[110]
p53 pDNA	Bortezomib	Cancer cell death	NCI-H1299	[111]

## 5. Perspective and Conclusions

The use of MSNs as gene delivery carriers in cancer therapy has shown promising results in preclinical studies. However, several challenges remain to be addressed to translate this technology into clinical practice. One important challenge is the development of efficient and safe surface modification strategies to enhance their stability, biocompatibility, and targeting ability. In the future, there is a need to further optimize the surface modification of MSNs for gene delivery, with a focus on the functionalization of the surface by targeting ligands or responsive moieties to enable specific cell uptake and controlled gene release. In addition, the combination of MSNs based gene delivery with other therapeutic modalities, such as chemotherapy or immunotherapy, can improve the efficacy of cancer therapy and reduce the risk of treatment resistance. Moreover, the translation of MSNs based gene delivery in the clinical setting requires a thorough understanding of the safety and toxicological profile of MSNs, including their long-term stability, biodistribution, and biodegradation in vivo. The development of appropriate pre-clinical models and regulatory guidelines is also essential for the successful translation of MSNs based gene delivery to become a new intervention.

In conclusion, the development of MSNs as a gene delivery carrier for cancer therapy is a promising area of research that requires further optimization and investigation. However, with the development of new and efficient surface modification strategies, and in-depth studies on the safety and toxicological profile of MSNs, this technology has the potential to significantly improve the efficacy and specificity of cancer therapy in the future.

## Figures and Tables

**Figure 1 pharmaceutics-15-01432-f001:**
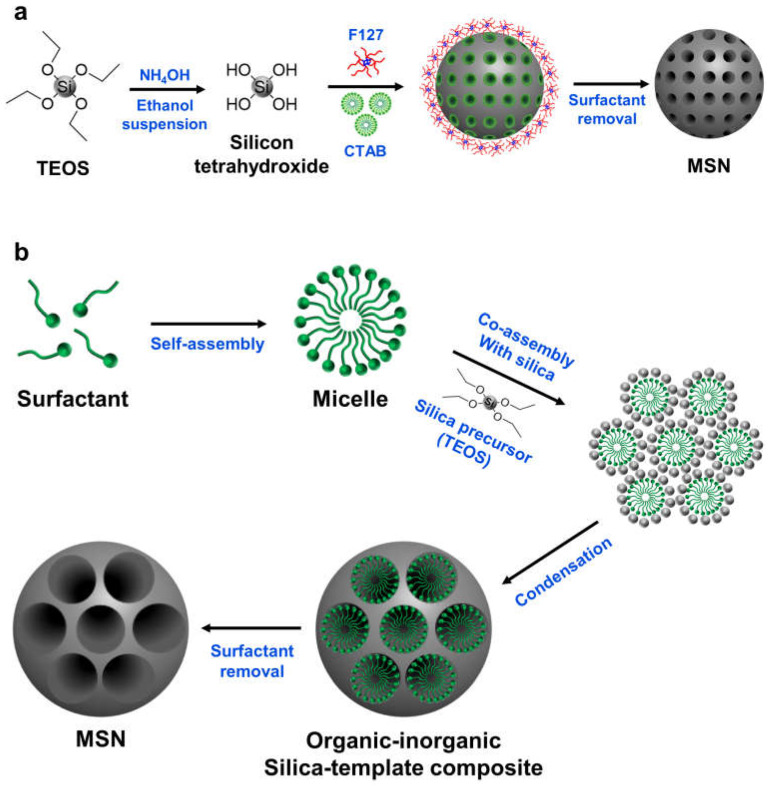
Schematic representation of mesoporous silica nanoparticle using different synthesis methods. (**a**) Stöber method for synthesis of mesoporous silica nanoparticles involves the hydrolysis and condensation of tetraethyl orthosilicate (TEOS) in the presence of surfactant templates cetyltrimethylammonium bromide (CTAB) and Pluronic F127 (F127) with ammonia and ethanol as a catalyst. NH_4_OH causes hydrolysis of TEOS, leading to the formation of Si-OH groups, which undergo condensation reactions to form Si-O-Si bonds and create the mesoporous structure. (**b**) Sol–gel method for synthesis of mesoporous silica nanoparticles involves hydrolysis of a silica precursor such as TEOS in the presence of a surfactant such as CTAB to form micelles. The hydrolyzed silica precursor polymerizes around the surfactant micelles to form a silica gel, which is then used to create the mesoporous structure with high surface area and tunable pore size guided by the surfactant micelles.

**Figure 2 pharmaceutics-15-01432-f002:**
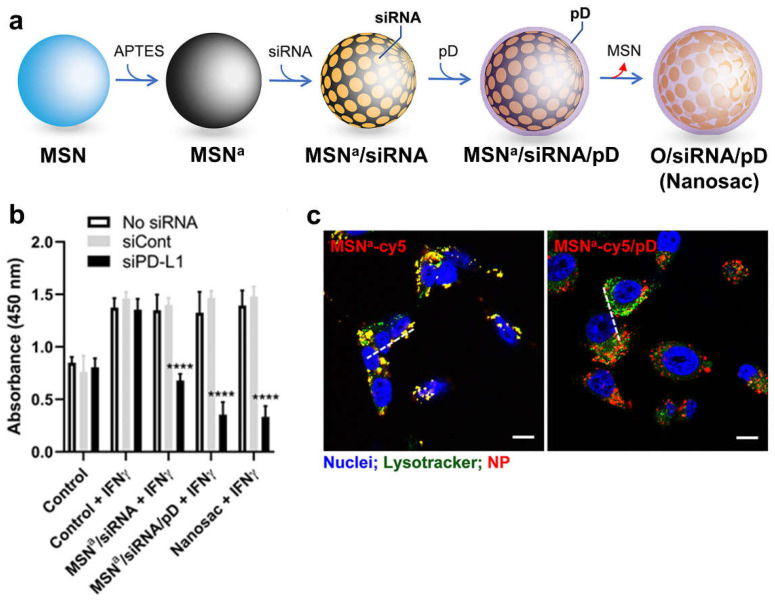
Effective gene silencing via polydopamine-coated MSNs loaded with PDL1 siRNA, which utilizes caveolae-mediated endocytosis for tumor cell entry. (**a**) Preparation of O/siRNA/pD (Nanosac). MSN^a^: MSN conjugated with APTES; MSN^a^/siRNA: siRNA loaded on MSN^a^; MSN^a^/siRNA/pD: polydopamine (pD) coated MSN^a^/siRNA; and O/siRNA/pD (Nanosac): siRNA-loaded nanocapsules. (**b**) Gene silencing by siPD-L1 in IFN-γ-activated CT26 cells. ****: *p* < 0.0001 vs. no siRNA by Dunnett’s multiple comparisons test following two-way ANOVA. (**c**) Confocal microscope images with cy5-labeled NPs. Scale bars: 10 μm. Reprinted/adapted with permission from [62]. Copyright 2021 American Chemical Society.

**Figure 3 pharmaceutics-15-01432-f003:**
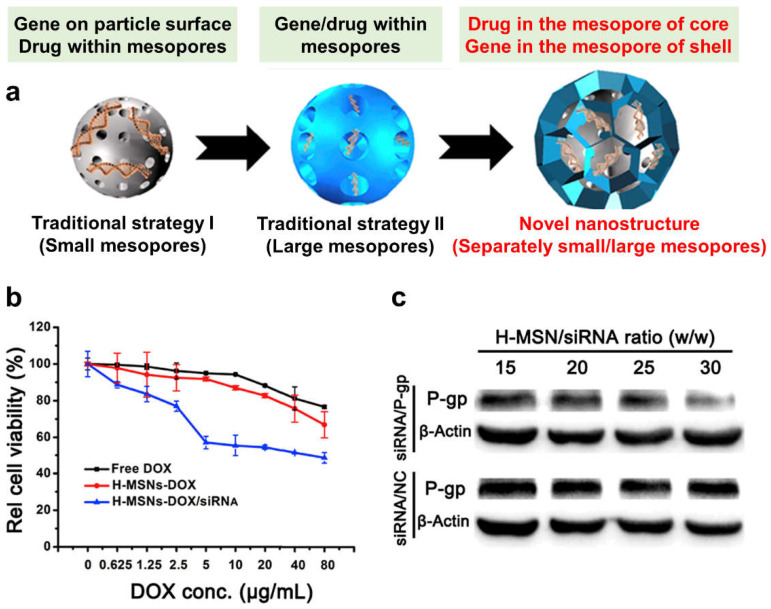
Core–shell hierarchical mesoporous silica/organosilica nanosystem with independent encapsulation of siRNA and doxorubicin through large and small mesopores in the shell and core, respectively. (**a**) Structures of small pore-sized MSNs, large pore-sized MSNs, and hierarchical MSNs (H-MSNs) for gene/drug co-delivery for MDR reversing. (**b**) MTT assay of MCF-7/ADR cancer cells after incubations with NPs for 48 h. (**c**) P-gp expressions of MCF-7/ADR cancer cells after the treatments with NPs (NC: negative control gene) at mass ratios of H-MSNs to siRNA (15, 20, 25, and 30). Reprinted/adapted with permission from [80]. Copyright 2017 Elsevier.

**Figure 4 pharmaceutics-15-01432-f004:**
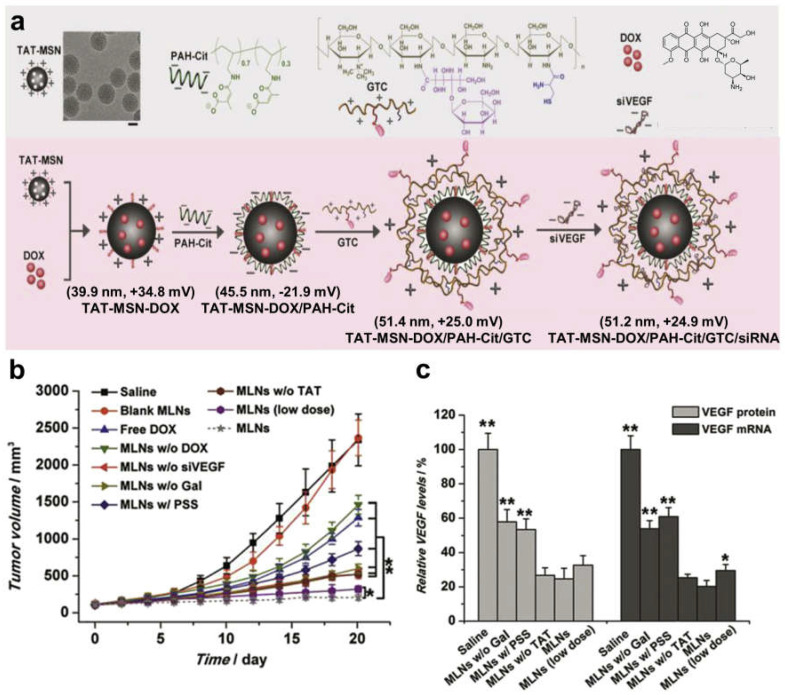
Efficient co-delivery of doxorubicin and VEGF siRNA achieved through multi-layered nanocomplexes (MLNs). (**a**) Schematic structures of MLNs which were constructed via layer-by-layer self-assembly driven by the electrostatic coverage of PAH-Cit and GTC onto the TAT-MSN core. Scale bar: 20 nm. (**b**) In vivo antitumor efficacy of MLNs. * *p* < 0.05, ** *p* < 0.01. (**c**) Relative VEGF protein and mRNA level in tumor tissue. * *p* < 0.05, ** *p* < 0.01 vs. VEGF levels after treatment with MLNs. Reprinted/adapted with permission from [92]. Copyright 2015 Elsevier.

## Data Availability

Not applicable.

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
