# Peer review of "Mesoporous Silica Nanoparticles as a Gene Delivery Platform for Cancer Therapy"

_pharmaceutics, 2023, doi:10.3390/pharmaceutics15051432_

Round 1

Reviewer 1 Report

Timely and well written review and I have only limited comments:

There are a lot of duplicate statements on the properties required of MSNs. E.g., phrasing in lines 51–54 repeated in slightly modified form for instance on line 128 and 148 (possibly elsewhere too). These could be rephrase to “required properties” or “favourable properties”

Figure 1 legend could do with a bit more explanation, also of the used abbreviations. 

Please use doxorubicin rather than Dox. It avoids confusion with doxycycline. 

Although siRNA has been used routinely, I do not think that this is nowadays considered a likely application. If MSNs can indeed deliver cargo to the tumor, more modern gene editing technology would likely be favoured. 

Please harmonise fonts (color, size, formatting) within the figures. 

There is a lot of original data in the manuscript (Figures 2, 3, and 4). I would propose to reduced this. 

Line 249; please rephrase. 

Table 1; can the authors attempt to fit the entries on a single line?

Lines 356 and 357; this a gross simplification of the rationale for combined therapies (there are many more hallmarks to target) and partly incorrect (many combination therapies are designed with patient heterogeneity in mind, not parallel biological processes). 

It would help the reader if all sections end with a concluding remark to summarise the (sometimes a bit anecdotal) listed findings. Also, a transition to the next section would be nice. 

Reviewer 2 Report

The paper reports a series of works about the use of mesoporous silica nanoparticles (MSN) as vehicles for molecules applied to cancer therapy, which is an important research.

The authors did not report any comparison on the molecule´s sizes and MSN pore size. For example, on line 133-135 they must provide the sizes of DNA and siRNA in order to make clear where the encapsulation occurs, that means, inside the mesopores or in the particles macroporosity and surface. 

It is very important to include in the manuscript a paragraph about this question, because non-porous silica particles could also be used with similar efficiency. In other words, the authors must to explain why the mesoporosity is important in MSN cancer therapy.

Reviewer 3 Report

I cannot find faults in the manuscript, so I congratulate the authors for their work and wish them the best.

Reviewer 4 Report

The manuscript entitled mesoporous silica nanoparticles as gene delivery platform for cancer therapy focuses on the anticancer possibilities of gene therapy employing such silica particles. Despite the manuscript is well written and structured and provides clear information on visited topics, to my opinion it also lacks on several important aspects of gene delivery that must be addressed before publication.

Regarding the scientific aspects, I would appreciate some deeper discussion on the following topics, that to my opinion have been significantly overlooked:

To provide useful information for the reader, I recommend the authors to include a brief section dealing with the most recent advances on preserving MSNs integrity against amine-promoted dissolution in aqueous media, which is of special interest for two-drug delivery strategies.

I consider the It would be also interesting to expand section 3 with the relationships between size and morphology that must have MSNs to provide best uptake rates. Moreover, it must be also discussed the advantages and limitations of in-pore loading and surface deposition strategies.

To conclude with scientific aspects, I agree with authors that combination therapies are one of the most promising fields of research for nanomedicine. However, I wonder why have the authors did not cover the use of single-nucleotide delivery? Is it because gene delivery alone does not offer satisfactory therapeutic effects? In any case, if combination therapy is covered with conventional chemotherapeutics, I would also appreciate a discussion on if two different RNAs may improve single-RNA therapies.

To conclude with the manuscript and bibliography provided, I suggest updating cited references, as only 33% of given references (41/123) are in the period within 2018-2022. To my opinion reported record must be significantly updated with special emphasis on recent publications. Moreover, many references (1, 8, 10, 12, 18, 16, 20, 22, 29-32, …) are incomplete with missing page numbers. These issues must be also corrected prior to publication.

Manuscript is well written and possible mistakes do not interfere with readability and undertanding of covered topics.

To my opinion no significant improvements are required on language editing.

Round 2

Reviewer 4 Report

Thank you for updating the manuscript and include claimed suggestions. Congratulations.